# Effects of Regorafenib, a Multi-Kinase Inhibitor, on Conjunctival Scarring in a Canine Filtration Surgery Model in Comparison with Mitomycin-C

**DOI:** 10.3390/ijms21010063

**Published:** 2019-12-20

**Authors:** Emika Nemoto, Shota Kojima, Tetsuya Sugiyama, Denan Jin, Shinji Takai, Michiko Maeda, Ryohsuke Kohmoto, Mari Ueki, Hidehiro Oku, Tsunehiko Ikeda

**Affiliations:** 1Department of Ophthalmology, Osaka Medical College, Takatsuki-City, Osaka 569-8686, Japan; emika0808@yahoo.co.jp (E.N.); tsugiyama@osaka-med.ac.jp (T.S.); opt182@osaka-med.ac.jp (M.M.); opt183@osaka-med.ac.jp (R.K.); opt025@osaka-med.ac.jp (H.O.); tikeda@osaka-med.ac.jp (T.I.); 2Department of Innovative Medicine, Osaka Medical College, Takatsuki-City, Osaka 569-8686, Japan; pha012@osaka-med.ac.jp (D.J.); pha010@osaka-med.ac.jp (S.T.); 3NAGATA Eye Clinic, Nara-City, Nara 631-0844, Japan; opt089@osaka-med.ac.jp

**Keywords:** beagles, glaucoma, multi-kinase inhibitor, regorafenib, trabeculectomy

## Abstract

Regorafenib eye drops were developed for treating age-related macular degeneration. This study aimed to investigate the effects of this multi-kinase inhibitor on intraocular pressure (IOP), bleb formation, and conjunctival changes in a canine filtration surgery model. Glaucoma filtration surgery models were created in 24 eyes of 24 beagles. In experiment 1 (Ex 1), regorafenib eye drops (regorafenib group: *n* = 6) or a vehicle (control group, *n* = 6) were instilled twice daily for 4 weeks postoperatively. In experiment 2 (Ex 2), regorafenib eye drops were instilled as in Ex 1 (regorafenib group: *n* = 6) for 12 weeks while conventional intraoperative mitomycin-C (MMC) was utilized (MMC group: *n* = 6), In Ex 1, only the regorafenib group showed significant IOP reduction with a significantly higher bleb score. Subconjunctival area, collagen density, vessels, and cells showing proliferation and differentiation were lower in subconjunctival tissue in the regorafenib group. In Ex 2, no significant difference was found in IOP reduction and bleb formation between the regorafenib and MMC groups; bleb walls were significantly thicker and collagen density and vessels were higher in the regorafenib group; and no differences were observed in the above-mentioned cells. Thus, regorafenib might be a better alternative to MMC for creating thicker and less ischemic blebs in glaucoma filtration surgery.

## 1. Introduction

Glaucoma is one of the major causes of irreversible blindness worldwide and is caused by retinal ganglion cell death, which leads to progressive visual field loss and blindness [1]. Intraocular pressure (IOP) is one of the main glaucoma risk factors, and its reduction is thought to be the most effective approach for suppression of glaucoma progression [2]. When medical and laser treatment fail to achieve the desired results, glaucoma filtration surgery (i.e., trabeculectomy), which involves draining the aqueous humor to the subconjunctival space and forming a filtration bleb, is chosen to reduce IOP [3]. However excessive postoperative wound healing and scarring with subsequent fibrosis may lead to the formation of smaller blebs or bleb disappearance, which is often the cause of surgical failure [4,5]. Since the early 1980s, antimetabolites such as mitomycin-C (MMC) have been used to lessen excessive wound healing after the surgery [6]. MMC has been shown to be a successful antifibrotic agent and improve the results of glaucoma filtration surgery in multiple large-scale clinical trials [7,8,9]. However, the application of MMC also increases the risk of complications such as a thin bleb, avascular bleb, bleb leakage, bleb infection, and infectious endophthalmitis in the late phase [10,11,12]. Therefore, an effective and safer agent to prevent failures of glaucoma filtration surgery and improve surgical outcomes is currently needed.

Platelet-derived growth factor (PDGF), vascular endothelial growth factor (VEGF), and transforming growth factor β (TGF-β) play important roles in wound healing [13]. The proliferating fibroblasts gradually differentiate into myofibroblasts in response to several factors, such as TGF-β [14]. VEGF induces TGF-β expression [15]. Myofibroblasts are characterized by expression of α-smooth muscle actin (αSMA) [16], which mediates wound contraction and the formation of a collagen-rich extracellular matrix. Excessive scar formation or fibrosis is characterized by the persistent presence of myofibroblasts [17]. Thus, inhibition of TGF-β, VEGF, and PDGF may be effective in suppressing wound healing. A number of studies have assessed the effects of these factors on glaucoma filtration surgery. In one trial, subconjunctival injections of anti-TGF-β antibody were administered as a substitute for MMC to suppress fibroblast proliferation post trabeculectomy, but the outcome was reportedly unsuccessful [18]. Clinical data indicate that VEGF antagonists are not superior to antimetabolites [19] but may serve as adjuncts to improve outcomes and reduce the need for secondary procedures [20,21]. Previous reports have suggested that PDGF stimulates Tenon’s capsule fibroblasts proliferation in vitro, and it plays an important role in the wound healing response after glaucoma filtering surgery [22,23]. However, these have not yet been applied to clinical treatment.

Regorafenib is a multi-kinase inhibitor of fibroblast growth factor receptor (FGFR); vascular endothelial growth factor receptor (VEGFR) 1–3; platelet-derived growth factor receptor (PDGFR); tyrosine kinase with immunoglobulin-like and EGF-like domain-2 (Tie-2); the stem cell factor receptor, c-KIT; rearranged during transfection (RET); and B-RAF [24,25]. It was approved by the Food and Drug Administration (FDA) for the treatment of colorectal cancer (CRC) in 2012, gastrointestinal stromal tumors (GIST) in 2013, and advanced hepatocellular carcinoma previously treated with Sorafenib in 2017 [25]. Previous reports [26] have investigated the effect of sunitinib, another multi-kinase inhibitor, on wound healing in the rabbit filtration surgery model. Sunitinib reduced fibroblasts and suppressed wound healing via the inhibition of TGF-β, FGF-β, and PDGF in the rabbit glaucoma filtration surgery model [26]. This indicates that regorafenib may have similar effects. Since regorafenib eye drops were developed as an anti-VEGFR therapy for the management of age-related macular degeneration [27,28], we used them in this study. The purpose of this study was to investigate the effects of regorafenib eye drops on conjunctival scarring in canine glaucoma filtration models.

## 2. Results

### 2.1. Experiment 1 (Ex 1): Comparison of the Effects in the Regorafenib and Control Groups

#### 2.1.1. IOP Change

The initial IOP values (mean ± SD) were 16.0 ± 3.6 mmHg in the regorafenib group and 16.2 ± 3.9 mmHg in the control group. The IOP values obtained 2 weeks postoperatively were 8.6 ± 1.9 mmHg in the regorafenib group and 11.5 ± 1.8 mmHg in the control group, while the corresponding values 4 weeks postoperatively were 11.4 ± 1.8 mmHg and 12.3 ± 2.1 mmHg, respectively. In the regorafenib group, IOP was significantly reduced at 2 and 4 weeks postoperatively (*p* < 0.05, paired *t*-test). In the control group, IOP showed a slight postoperative reduction, but it was not significant (Figure 1).

#### 2.1.2. Bleb Score

The bleb scores (mean ± SD) at 2 weeks postoperatively were 3.7 ± 0.5 in the regorafenib group and 2.7 ± 0.5 in the control group, while those at 4 weeks postoperatively were 2.5 ± 0.5 in the regorafenib group and 1.5 ± 0.5 in the control group. The bleb score was significantly higher in the regorafenib group than in the control group at 2 and 4 weeks postoperatively (*p* < 0.05, Mann–Whitney U test, Figure 1).

#### 2.1.3. Subconjunctival/Scleral Area Ratio

Figure 2 shows that the subconjunctival area in the regorafenib group was thinner than that in the control group. The ratio of the subconjunctival area to the scleral area was significantly lower in the regorafenib group than in the control group (*p* = 0.025, Mann–Whitney U test; Table 1).

#### 2.1.4. Density of Collagen in Subconjunctival Tissue

The collagen area identified by the color extraction method is shown in green (Figure 2). The density of collagen in subconjunctival tissue was significantly lower in the regorafenib group compared with that in the control group (Table 1).

#### 2.1.5. Density of Vessels in Subconjunctival Tissue

The density of vessels in subconjunctival tissue was significantly lower in the regorafenib group compared with that in the control group (Figure 3, Table 1).

#### 2.1.6. Vimentin-, TGF-β-, Proliferative Cell Nuclear Antigen (PCNA)-, and αSMA-Positive Cells

A lower number of vimentin-, TGF-β-, PCNA-, and αSMA-positive cells were found in the regorafenib group than in the control group (Figure 4, Table 1).

### 2.2. Experiment 2 (Ex 2): Comparison of the Effects in the Regorafenib and MMC Groups

#### 2.2.1. IOP Change

The initial IOP values (mean ± SD) were 15.9 ± 1.1 mmHg in the regorafenib group and 16.9 ± 1.2 mmHg in the MMC group. The IOP values obtained 4 weeks postoperatively were 12.6 ± 0.8 mmHg in the regorafenib group and 11.6 ± 2.0 mmHg in the MMC group; those obtained 8 weeks postoperatively were 11.6 ± 1.3 mmHg in the regorafenib group and 11.9 ± 0.9 mmHg in the MMC group; and those obtained 12 weeks postoperatively were 12.3 ± 0.4 mmHg in the regorafenib group and 12.8 ± 1.1 mmHg in the MMC group. IOP was found to be significantly reduced at 4, 8, and 12 weeks postoperatively (*p* < 0.05, repeated measures ANOVA) in both groups. There was no significant difference in IOP between the eyes at each measurement point (* *p* > 0.5, Mann–Whitney U test) (Figure 5).

#### 2.2.2. Bleb Score

The bleb scores (mean ± SD) obtained 4 weeks postoperatively were 3.6 ± 0.5 in the regorafenib group and 3.5 ± 0.5 in the MMC group; those obtained 8 weeks postoperatively were 3.5 ± 0.5 in the regorafenib group and 3.5 ±0.5 in the MMC group; and those obtained 12 weeks postoperatively were 3.1 ± 0.4 in the regorafenib group and 3.0 ± 0.0 in the MMC group. The bleb scores significantly increased until 12 weeks postoperatively in both groups (*p* < 0.05, repeated measures ANOVA), and there was no significant difference between the two groups at each measurement point (* *p* > 0.5, Mann–Whitney U test) (Figure 5).

#### 2.2.3. Ultrasonic Biomicroscope (UBM) Assessments

The bleb wall thickness (mean ± SD) was 1.0 ± 0.1 mm in the regorafenib group and 0.8 ± 0.1 mm in the MMC group. The wall in the regorafenib group was thicker than that in the MMC group (*p* < 0.05, Mann–Whitney U test) (Figure 6).

#### 2.2.4. Density of Collagen in Subconjunctival Tissue

The collagen area identified by the color extraction method is shown in green (Figure 7). The density of collagen in subconjunctival tissue was significantly lower in the MMC group compared with that in the regorafenib group. (Table 2)

#### 2.2.5. Density of Vessels in the Subconjunctival Tissue

The density of vessels in the subconjunctival tissue was significantly lower in the MMC group compared with that in the regorafenib group (Figure 8, Table 2).

#### 2.2.6. Vimentin-, TGF-β-, PCNA-, and SMA-Positive Cells

There was no significant difference in the numbers of these cells between the two groups (Figure 9, Table 2).

## 3. Discussion

This is the first study assessing the effect of regorafenib on wound healing after glaucoma filtration surgery. Beagles were utilized in this study because the corneoscleral trabecular meshwork and angular aqueous plexus in the canine angle are similar to the conventional outflow and uveoscleral outflow in humans [29]. We also used our previously described [30,31] simple sclerotomy as a filtration surgery model, in which the scleral flap and suturing flap associated with conventional trabeculectomy are not made. To precisely evaluate the effects of drugs in a surgery model, the outward aqueous flow in all experimental eyes must be the same. However, controlling suture tightness to obtain the same outward aqueous flow is challenging, and therefore, we deemed simple sclerotomy to be the most appropriate model of filtration surgery.

First, to investigate the effects of regorafenib in reducing IOP and maintaining bleb after glaucoma filtration surgery, we examined these changes during the first postoperative month in Ex 1. In the early stages of wound healing, most active reactions occur and crucial cytokines, such as TGF-β and VEGF, are released [4]. Therefore, it is important to elucidate the changes occurring in this term physiologically, morphologically, and histologically. In Ex 1, IOP did not change significantly in the simple filtration group at 4 weeks, which is similar to the findings of a previous study [30]. We found that IOP was significantly reduced at 2 and 4 weeks postoperatively in the regorafenib group, and the bleb scores at 2 and 4 weeks postoperatively were significantly higher in the regorafenib group than in the control group. The change in bleb score did not completely reflect that in IOP, especially at 4 weeks postoperatively. This might be a result of either the scaring change within the bleb or the quality of slit lamp examination. In any case, regorafenib maintained the filtration bleb and promoted a prolonged reduction of IOP. We performed histological experiments to investigate the mechanism underlying the sustained bleb formation; we found that the ratio of the subconjunctival area to scleral area and the collagen density in subconjunctival tissue were significantly lower in the regorafenib group than in the control group, which indicates that the above-described bleb formation in the regorafenib group resulted from an inhibition of cell proliferation and fibrosis after glaucoma filtration surgery. To confirm this, we performed immunohistological staining, which revealed that the fibroblast densities (fibroblasts were labeled with anti-vimentin antibody) in the lesion were significantly lower in the regorafenib group than in the control group. The numbers of TGF-β-positive cells, PCNA-positive cells, and αSMA-positive cells were also significantly lower in the regorafenib group than in the control group. Taken together, these results verified our hypothesis. Regorafenib suppressed fibroblasts and also suppressed their differentiation into myoblasts by suppressing TGF-β. Regorafenib inhibits FGFR, PDGFR, and VEGFR [24,25], therefore, it might reduce fibroblast infiltration by inducing fibroblast apoptosis or suppressing the effect of FGF, PDGF, and VEGF.

As a result, regorafenib was effective in maintaining bleb formation and reducing IOP in 4 weeks. However, our findings raised some questions. First, the subconjunctival/scleral ratio and the density of collagen in subconjunctival tissue were significantly lower in the regorafenib group, indicating that regorafenib eye drops might cause the formation of thin bleb walls. Second, the density of vessels was lower in the regorafenib group, which is thought to be due to the inhibition of VEGFR. This indicates the formation of avascular blebs. Third, it was unclear whether these effects of regorafenib eye drops last as long as those of ordinary MMC use in trabeculectomy and are associated with fewer side effects. Thus, it was necessary to compare the effects of regorafenib and the conventional MMC method on IOP, bleb formation, and histological changes for the future clinical use.

In Ex 2, we found that IOP reduction and bleb score were not significantly different in the regorafenib and MMC groups. We also found that the densities of fibroblasts, TGF-β-positive cells, PCNA-positive cells, and αSMA-positive cells were not significantly different between the two groups. These data suggest that regorafenib eye drops have the same morphological and histological effects on IOP, bleb formation, and fibroblastic changes. However, the density of collagen and the vessels in subconjunctival tissue were higher in the regorafenib group than in the MMC group. Further, we used UBM to measure the thickness of the bleb walls, as reported previously [32,33], and found that bleb walls were thicker in the regorafenib group than in the MMC group. These results suggest that topical regorafenib after glaucoma filtration surgery may yield more favorable bleb formation in comparison with intraoperative MMC use. MMC inhibits the synthesis of DNA for cell growth and suppresses fibroblast proliferation [34]. MMC also reduces the secretion of collagen [35] and induces apoptosis in what appears to be a c-Jun N-terminal kinase 1 (NJK1) signaling-dependent mechanism [36]. Unfortunately, the effect of regorafenib on bleb formation has not been reported in detail, although regorafenib eye drops have been shown to significantly reduce choroidal neovascularization in a rat model through inhibition of VEGFR-2 [37]. Regorafenib also inhibits growth factors (such as PDGF, VEGF, and FGF) and cytokines (such as TGF-β) and suppresses the proliferation of fibroblasts. Due to inhibition along the cascade of wound healing, regorafenib may have a different and milder effect on subconjunctival tissue than MMC in relation to fibroblast death. These results suggest that regorafenib may be an alternative to MMC. Moreover, regorafenib may form thick walls and non-avascular blebs.

In comparison with a previous study [26] performed using sunitinib, which inhibits VEGFR, PDGFR, c-KIT, and RET, regorafenib also inhibited fibroblasts and TGF-β and prevented bleb scarring. Regorafenib may be more useful because it inhibits more kinases such as FGFR than sunitinib. Moreover, the safety of regorafenib eye drops has been demonstrated in clinical trials for age-related macular degeneration [27]. Side effects such as hand-foot syndrome and hypertension are problems with multi-kinase inhibitors. The absolute systemic exposure to regorafenib after multiple administrations of regorafenib eye drops 0.75 mg (one drop bid or tid, one eye treated) was 600–700 fold lower than that after oral administration of 160 mg, the approved dose for use in colorectal cancer/gastrointestinal stromal tumor, so the risk of side effects is expected to be low. Eyelid redness and edema, conjunctival hyperemia, and edema were observed across all regorafenib eye drop cohorts, but these were mild and comparable with the effect of placebo. On the basis of the above findings and the fact that eye drops have already been developed, we believe that regorafenib is easier to apply clinically than sunitinib.

It should be noted that there were several limitations in the present study. First, we did not set a control group in Ex 2, a long-term comparison. Because we have already indicated that the control group showed no IOP reduction and bleb maintenance in Ex 1, a short-term comparison, it is also important to reduce the number of sacrificed beagles in the Ex 2. Second, since the simple sclerectomy used in the study might have a different effect on growth factors and cytokine expression around the flap site, careful interpretation from this experiment is necessary. Third, elucidation of the detailed mechanism of suppression of wound healing by regorafenib is needed. Fourth, we need to investigate the optimal concentration for glaucoma filtering surgery because the concentration of eye drops used in this study is typically used for the treatment of age-related macular degeneration. Furthermore, it is necessary to investigate the period of instillation after surgery.

## 4. Materials and Methods

### 4.1. Drugs

Regorafenib eye drops (2%) and vehicle (100% paraffin) eye drops were provided by Bayer AG (Berlin, Germany). MMC was purchased from Kyowa Kirin Co., Ltd. (Tokyo, Japan).

### 4.2. Animals and IOP Measurements

The Committee of Animal Use and Care of Osaka Medical College approved the experimental protocols used in this study (No. 30099, 25 July 2018). This study involved 24 eyes of 24 beagles purchased from Japan SLC, Inc. (Hamamatsu, Japan). The beagles were housed in an air-conditioned room at approximately 23 °C and 60% humidity with a 12-h light–dark cycle, were fed regular canine food, and had constant free access to tap water. All of the animal experiments were conducted in accordance with the ARVO Statement for Use of Animals in Ophthalmic and Vision Research. The IOP measurements were obtained using a TONO-PEN AVIA^®^ VET (Reichert, Depew, NY, USA) in a front-facing position under general anesthesia with intermuscular injection of midazolam (0.3 mg per kg body weight) and medetomidine (0.02 mg per kg body weight) and intravenous injection of pentobarbital sodium (5–10 mg per kg body weight).

### 4.3. Glaucoma Filtration Surgery Model

The beagles were anesthetized as described above. The glaucoma filtration surgery model was created as previously described [30,31]. Briefly, a control suture was first fixed to the cornea using 8-0 Vicryl^®^ (Ethicon US, LLC., Dallas, TX, USA) sutures. Next, a 10 mm fornix-based flap of conjunctiva and the Tenon’s capsule (length, 5 mm) was made. After a 3 × 1 mm scleral portion was removed at the limbus, peripheral iridectomy was performed, followed by closing the conjunctiva with a 10-0 nylon suture. After surgery, the appropriate amount of 1 cm of 3 mg/g ofloxacin ointment was applied to the eye.

### 4.4. Experiment Protocol

#### 4.4.1. Experiment 1: Comparison of the Effects in the Regorafenib and Control Groups

To investigate the effect of regorafenib in the early stages of wound healing after glaucoma surgery, 12 eyes of 12 beagles were used. One drop (50 μL) of 2% regorafenib eye drop (regorafenib group, *n* = 6) or the vehicle (100% paraffin) (control group, *n* = 6) was instilled twice daily for 4 weeks after the surgery. IOP and bleb scores were assessed every 2 weeks for 4 weeks postoperatively, followed by histological evaluation.

#### 4.4.2. Experiment 2: Comparison of the Effects in the Regorafenib and MMC Groups

To compare the effect of regorafenib with MMC after glaucoma surgery, a longer observation period than that used in Ex 1 was necessary to evaluate future clinical use. In this experiment, 12 eyes of 12 beagles were used. Six eyes of 6 beagles were instilled with 50 μL of 2% regorafenib eye drops (regorafenib group) twice daily for 12 weeks after surgery. In the MMC group (*n* = 6), a sponge (Material Quick Absorber (MQA); Inami & Co., Tokyo, Japan), soaked with 0.04% MMC solution in distilled water was placed under the conjunctiva for 5 min before opening the scleral portion intraoperatively. IOP and bleb scores were assessed at 4, 8, and 12 weeks postoperatively. At 12 weeks, the thickness of the bleb wall was measured using ultrasonic biomicroscopy (UBM) and histological evaluations were performed.

### 4.5. Bleb Scores

Slit-lamp microscopy was used to examine and grade the blebs according to the definition previously reported by Perkins et al. [38], which reflects increasing bleb height and size as follows: 1, minimally high conjunctival thickening without swelling; 2, mild swelling present; 3, elevated bleb covering an area equivalent to 2–3 clock hours of the eye; and 4, greatly elevated bleb covering an area equivalent to more than 4 clock hours of the eye. A score of 0 indicated no observed bleb.

### 4.6. Ultrasound Biomicroscopy (UBM)

To measure the thickness of the bleb walls 12 weeks postoperatively, UBM (UD-800 Ultrasonic A/B Scanner and Biometer; Tomey Corporation, Nagoya, Japan) was used. We observed blebs in a front-facing position under the general anesthesia described above. In each scan, at the maximum height point of the bleb, a line from the top of the bleb wall to the bottom was drawn vertically and device’s software automatically calculated the maximum thickness of the bleb. We obtained measurements five times and averaged them for each eye.

### 4.7. Histological Examination

After the final measurement, beagles were euthanized with a lethal dose of KCl injected intracardially under general anesthesia. Ophthalmectomy was subsequently performed. The bleb area was identified with a marked 10-0 nylon suture and a 10 × 5 mm region including the conjunctiva, subconjunctival tissue, and sclera was excised. Conjunctival and scleral tissue specimens were fixed for 24 h with Carnoy Solution (Muto Pure Chemicals Co., Ltd., Tokyo, Japan), and embedded in paraffin for histologic analysis. Five-µm-thick sections were cut and mounted on silanized slides (Dako, Glostrup, Denmark), and subsequently deparaffinized with xylene and a series of graded ethanol solutions. Azan-Mallory staining was used to identify collagen fibers. The conjunctival and scleral areas of the lesion where the flap was made were assessed by using a computerized morphometry system (MacSCOPE Ver 2.2; Mitani Corporation, Fukui, Japan), and the ratio of the conjunctival area to scleral area was then calculated. A computerized color extraction system (Win Roof Ver. 6.13; Mitani Corporation) was used to quantify the collagen density in the conjunctiva by calculating the ratio of the Azan-Mallory stained collagen to the total conjunctival area. Mast cells were stained with Toluidine Blue for identification.

For antigen retrieval, sections were pretreated with 10 mM citrate buffer at pH 6.0, and subsequently autoclaved for 5 min at 120 °C before immunohistochemical staining. To remove endogenous peroxidase activity, the sections were then soaked in absolute methanol containing 3% hydrogen peroxide for 5 min at room temperature. To suppress nonspecific binding, the sections were incubated with Serum-Free Protein Block (X0909; Dako, Glostrup, Denmark) for 5 min. To label fibroblasts and vessels, anti-bovine vimentin antibody (Wako Pure Chemical Industries, Ltd., Osaka, Japan), mouse antihuman von Willebrand factor antibody (Wako Pure Chemical Industries, Ltd., Osaka, Japan), respectively, and antihuman α-SMA antibody (Dako, Glostrup, Denmark) were used. The sections were incubated overnight at 4 °C with each antibody, followed by reaction with the appropriate reagents from a streptavidin-biotin peroxidase kit (Dako Glostrup, Denmark) and 3-amino-9-ethylcarbazole for 5 to 10 min. The sections were subsequently counterstained with hematoxylin. The mouse monoclonal antibody against PCNA (PC10) (M0879; Glostrup, Denmark) was used to label PCNA-positive cells. Polyclonal Chicken IgY anti-TGF-β1-2 antibody (R&D Systems, Inc., Minneapolis, MN, USA) was used to label TGF-β-positive cells. The number of vimentin-positive cells, TGF-β-positive cells, PCNA-positive cells, and αSMA-positive cells as well as vessel density were counted in the conjunctival and scleral lesions using a light microscope (number per ×100 field), and the average number of each type of cells or vessels in five randomly selected fields was then calculated.

### 4.8. Masking

All measurements were performed by investigators (EN and SK) who were masked from identifying specific treatments.

### 4.9. Statistical Analysis

Each measurement was expressed as the mean ± SD. Repeated-measures ANOVA was used for statistical comparisons of repeated measurements, followed by other tests. IOP was statistically analyzed with a paired *t*-test in comparison with the initial values in the two groups. Other parameters were evaluated via the Mann–Whitney U test. Differences were considered statistically significant at a *p*-value of < 0.05.

## 5. Conclusions

Topical application of regorafenib maintained bleb formation and reduced IOP to a similar degree to for MMC. Moreover, the effect on conjunctiva was less than that of MMC. These results suggest that regorafenib may be an alternative to MMC.

## Figures and Tables

**Figure 1 ijms-21-00063-f001:**
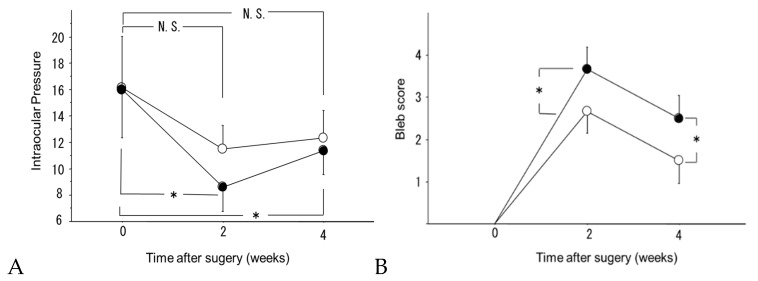
(**A**) The effect of regorafenib eye drops on IOP changes: IOP changes in the control group (○) and in the regorafenib group (●). Data are shown as the mean ± SD for 6 beagles. (* *p* < 0.05, paired *t*-test. N.S: *p* ≥ 0.05, paired *t*-test.). IOP reduction was maintained in the regorafenib group while no significant change was observed in the control group. (**B**) Comparison of bleb score. Bleb score changes in the control group (○) and in the regorafenib group (●). Data are shown as the mean ± SD for 6 beagles. The bleb score was significantly higher in the regorafenib group than in the control group at 2 and 4 weeks postoperatively (* *p* < 0.05, Mann–Whitney U test).

**Figure 2 ijms-21-00063-f002:**
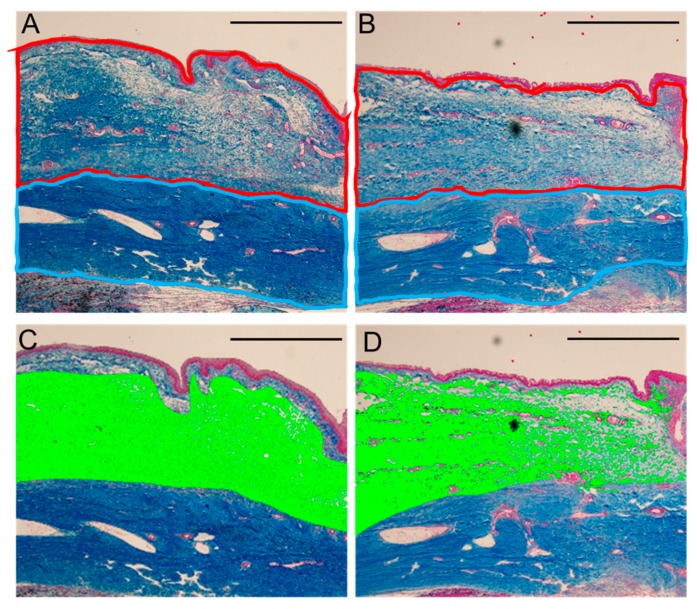
Representative photomicrographs of the conjunctiva obtained from the eyes treated in the control group (**A**,**C**) and in the regorafenib group (**B**,**D**) at 4 weeks postoperatively and stained with azan stain. The collagen fibers are stained blue. The subconjunctival and scleral areas are surrounded by red and light-blue lines, respectively (**A**,**B**). The area in green identified by use of the color extraction method illustrates collagen in the conjunctiva (**C**,**D**). Scale bars: 1000 μm.

**Figure 3 ijms-21-00063-f003:**
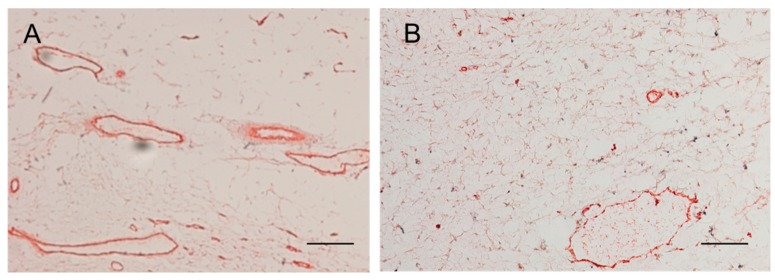
The vessels in subconjunctival tissue in the control group (**A**) and the regorafenib group (**B**) at 4 weeks postoperatively. Scale bars: 100 μm.

**Figure 4 ijms-21-00063-f004:**
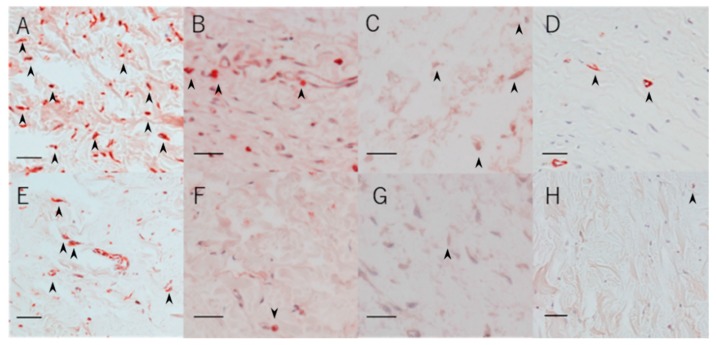
Representative immunohistochemical staining images of the sections for vimentin-positive cells in the control group (**A**) and the regorafenib group (**E**), for TGF-β-positive cells in the control group (**B**) and the regorafenib group (**F**), for PCNA-positive cells in the control group (**C**) and the regorafenib group (**G**), and for αSMA-positive cells in the control group (**D**) and the regorafenib group (**H**). Cells are indicated by black arrows. Scale bars: 50 μm.

**Figure 5 ijms-21-00063-f005:**
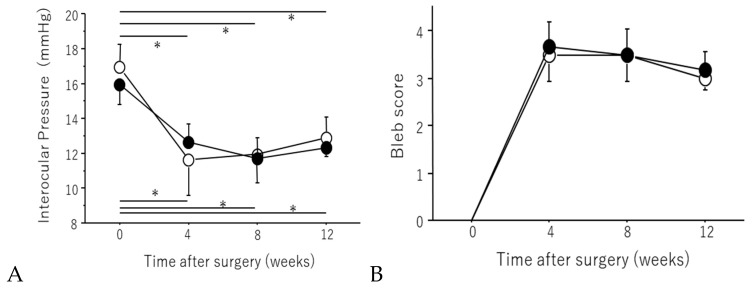
(**A**) The effects on IOP changes. IOP changes in the regorafenib group (●) and the MMC group (○). Data are shown as the mean ± SD for 6 beagles. IOP was found to be significantly reduced at 4, 8, and 12 weeks postoperatively in both groups (* *p* < 0.05, repeated-measures ANOVA). There was no significant difference in IOP between the eyes at each measurement point (*p* > 0.5, Mann–Whitney U test). (**B**) Comparison of bleb score. Bleb score changes in the regorafenib group (●) and the MMC group (○). Data are shown as the mean ± SD for 6 beagles. The bleb score significantly increased until 12 weeks postoperatively in both groups (*p* < 0.05, repeated measures ANOVA), and there was no significant difference between the two groups at each measurement point (* *p* > 0.5, Mann–Whitney U test).

**Figure 6 ijms-21-00063-f006:**
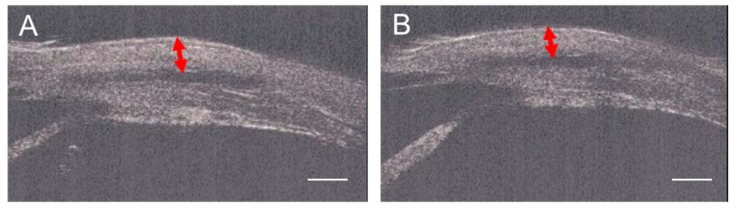
UBM image of the bleb wall (indicated by arrows) in the regorafenib group (**A**) and in the MMC group (**B**). The wall in the regorafenib group was thicker than that in the MMC group (*p* < 0.05, Mann–Whitney U test). Scale bars: 1 mm.

**Figure 7 ijms-21-00063-f007:**
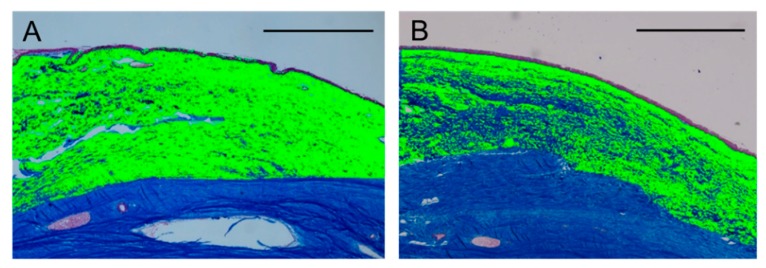
Representative photomicrographs of the conjunctiva obtained from eyes treated in the regorafenib group (**A**) and in the MMC group (**B**) at 12 weeks postoperatively and stained with azan stain. The area in green identified by the color extraction method illustrates collagen in the conjunctiva. Scale bars: 1000 μm.

**Figure 8 ijms-21-00063-f008:**
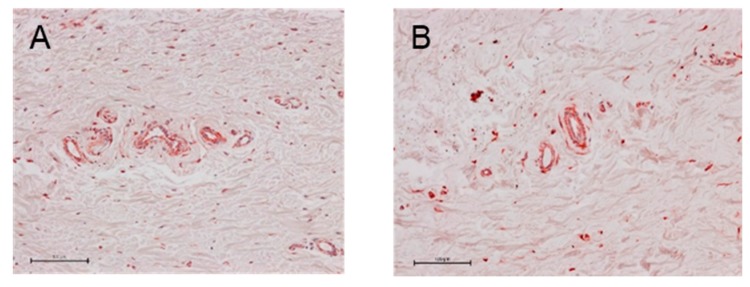
The capillaries of subconjunctival tissue in the regorafenib group (**A**) and the MMC group (**B**) at 12 weeks postoperatively. Scare bars: 100 μm.

**Figure 9 ijms-21-00063-f009:**
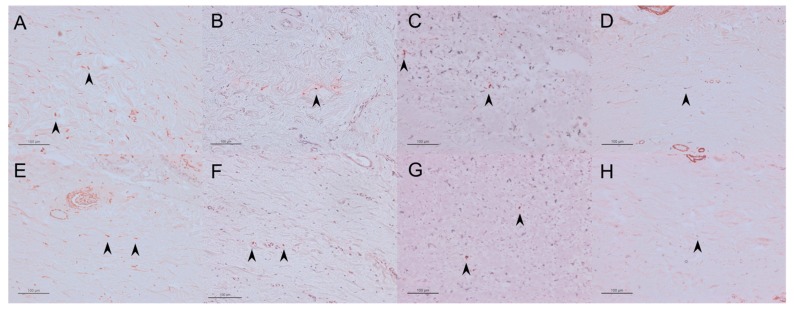
Representative immunohistochemical staining images of the section for vimentin-positive cells in the MMC group (**A**) and the regorafenib group (**E**), for TGF-β-positive cells in the MMC group (**B**) and the regorafenib group (**F**), for PCNA-positive cells in the MMC group (**C**) and the regorafenib group (**G**), and for αSMA-positive cells in the MMC group (**D**) and the regorafenib group (**H**). Cells are indicated by black arrows. Scale bars: 100 μm.

**Table 1 ijms-21-00063-t001:** Comparison of the ratio of the conjunctival area to the scleral area, collagen density of subconjunctival tissue, density of vessels, vimentin-positive cells, TGF-β-positive cells, PCNA-positive cells, and αSMA-positive cells in subconjunctival tissue between the control and regorafenib groups. Data are shown as the mean ± SD for 6 eyes of 6 beagles.

Index	Control Group	Regorafenib Group	*p*-Value (Mann–Whitney U)
Ratio of the conjunctival area to the scleral area	1.1 ± 0.1	0.8 ± 0.2	0.025
Collagen density of subconjunctival tissue, %	89.3 ± 5.9	75.5 ± 9.1	0.025
Density of vessels in subconjunctival tissue, per·mm^2^	14.1 ± 4.7	8.3 ± 4.0	0.03
Density of vimentin-positive cells, per·mm^2^	33.8 ± 15.4	10.3 ± 9.3	0.02
Density of TGFβ-positive cells, per·mm^2^	18.3 ± 5.7	5.6 ± 3.9	0.007
Density of PCNA-positive cells, per·mm^2^	13.8 ± 6.8	6.3 ± 3.0	0.045
Density of αSMA-positive cells, per·mm^2^	3.3 ± 1.3	0.3 ± 0.5	0.004

**Table 2 ijms-21-00063-t002:** Comparisons of the ratio of the conjunctival area to the sclera area, collagen density of subconjunctival tissue, density of vessels in subconjunctival tissue, and densities of vimentin-positive cells, TGF-β-positive cells, PCNA-positive cells, and αSMA-positive cells between the regorafenib group and the MMC group. Data are shown as the mean ± SD for 6 eyes of 6 beagles.

Index	Regorafenib Group	MMC Group	*p*-Value (Mann–Whitney U)
Ratio of the conjunctival area to the scleral area	1.5 ± 0.4	1.4 ± 0.3	0.45
Collagen density of subconjunctival tissue, %	82.5 ± 8.1	62.3 ± 17.6	0.01
Density of vessels in the subconjunctival tissue, per·mm^2^	11.5 ± 2.0	6.5 ± 1.9	0.01
Density of vimentin-positive cells, per·mm^2^	5.5 ± 0.7	5.0 ± 0.2	0.63
Density of TGFβ-positive cells, per·mm^2^	3.8 ± 0.9	3.5 ± 1.2	0.63
Density of PCNA-positive cells, per·mm^2^	6.3 ± 1.3	5.3 ± 1.2	0.2
Density of αSMA-positive cells, per·mm^2^	1.1 ± 0.9	1.0 ± 0.8	0.87

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
