# Peer review of "Effects of Regorafenib, a Multi-Kinase Inhibitor, on Conjunctival Scarring in a Canine Filtration Surgery Model in Comparison with Mitomycin-C"

_ijms, 2019, doi:10.3390/ijms21010063_

Round 1

Reviewer 1 Report

In this paper, authors have undertaken a series of experiments and shown that the regorafenib, a multi kinase inhibitor approved for several cancers, might be a better alternative to mitomycin-C on conjunctival scarring in glaucoma filtration surgery. The methodology is appropriate, results clear and conclusions appropriate. I think this manuscript need only slight modifications to publish.

In figure 1, bleb score four weeks after surgery is significantly different between control group and regorafenib group. However, the difference of bleb score is not reflected IOP at this time point. Author should explain the reason. Is the difference caused by quality of bleb not detected by slit lamp examination?

Authors described the risk of MMC such as thin bleb, avascular bleb, bleb leakage, bleb infection, and infectious endoophthalmitis in the late phase. In this study, advantage of regorafenib on some of these complication are clearly presented. If there are any data, report, or speculation, describe them and discuss about the difference between MMC and regorafenib on bleb leakage and infection.

“data are shown as the mean ± SD for 12 beagles” in figure/table legends should correct to “data are shown as the mean ± SD for 6 beagles”, because each data point are consisted from six animals per group.

Author Response

We really appreciate the Reviewer’s positive comments regarding our manuscript. We have now been encouraged to make progress in our experiments for the treatment of glaucoma. In the revised manuscript, the sections highlighted with yellow were identified by the journal as being similar to text from previous publications; therefore, we have also rewritten this text. We have used the "Track Changes" function in Microsoft Word to highlight our revisions, as suggested, so that the changes are easily visible to the editors and reviewers.

Point 1: In figure 1, bleb score four weeks after surgery is significantly different between control group and regorafenib group. However, the difference of bleb score is not reflected IOP at this time point. Author should explain the reason. Is the difference caused by quality of bleb not detected by slit lamp examination?

Response 1: Thank you for your constructive comment. We have added the following evaluation of the result to the discussion part. “The change in bleb score did not completely reflect that in IOP, especially at 4 weeks postoperatively. This might be a result of either the scaring change within the bleb or the quality of slit lamp examination. “(Page 8, Line 232-235).

Point 2: Authors described the risk of MMC such as thin bleb, avascular bleb, bleb leakage, bleb infection, and infectious endoophthalmitis in the late phase. In this study, advantage of regorafenib on some of these complication are clearly presented. If there are any data, report, or speculation, describe them and discuss about the difference between MMC and regorafenib on bleb leakage and infection.

Response 2: Thank you for your constructive comment. We have added the following text to compare the effect of MMC and regorafenib on bleb formation to the discussion part. “MMC also reduces the secretion of collagen [35] and induces apoptosis in what appears to be a c-Jun N-terminal kinase 1 (NJK1) signaling-dependent mechanism [36] Unfortunately, the effect of regorafenib on bleb formation has not been reported in detail, although regorafenib eye drops have been shown to significantly reduce choroidal neovascularization in a rat model through inhibition of VEGFR-2 [37]. “(Page 8, Line 269- Page 9 Line 274). We have also added “also” in Page 9 Line 274 and “different and” in Line 276.

Point 3: “data are shown as the mean ± SD for 12 beagles” in figure/table legends should correct to “data are shown as the mean ± SD for 6 beagles”, because each data point are consisted from six animals per group.

Response 3: Thank you for your constructive comment. We apologize for this mistake and are grateful you have taken the time to check the manuscript. We appreciate your kind comments and have corrected all the numbers accordingly.

Reviewer 2 Report

Comments:

The manuscript "Effects of regorafenib, a multi-kinase inhibitor, on conjunctival scarring in a canine filtration surgery model in comparison with mitomycin-C" by Nemoto et al presents interesting findings that suggest the ability of Regorafenib  to be a better alternative to Mitomycin‑C for creating thicker and lesser ischemic blebs in glaucoma filtration surgery. In general, the manuscript is well-written, and the results support the hypothesis.

             However, the discussion section should be elaborated to include:

previous studies that showed the effects of Mitomycin C on bleb formation and related factors in trabeculectomy; and literature that demonstrates the detailed effects of Regorafenib in the eye.

Author Response

We really appreciate the Reviewer’s positive comments regarding our manuscript. We have now been encouraged to make progress in our experiments for the treatment of glaucoma. In the revised manuscript, the sections highlighted with yellow were identified by the journal as being similar to text from previous publications; therefore, we have also rewritten this text. We have used the "Track Changes" function in Microsoft Word to highlight our revisions, as suggested, so that the changes are easily visible to the editors and reviewers.

Point 1: However, the discussion section should be elaborated to include: previous studies that showed the effects of Mitomycin C on bleb formation and related factors in trabeculectomy; and literature that demonstrates the detailed effects of Regorafenib in the eye.  ?

Response 1: Thank you for your constructive comment. We have added the following text to compare the effect of MMC and regorafenib on bleb formation to the discussion part. “MMC also reduces the secretion of collagen [35] and induces apoptosis in what appears to be a c-Jun N-terminal kinase 1 (NJK1) signaling-dependent mechanism [36] Unfortunately, the effect of regorafenib on bleb formation has not been reported in detail, although regorafenib eye drops have been shown to significantly reduce choroidal neovascularization in a rat model through inhibition of VEGFR-2 [37]. “(Page 8, Line 269- Page 9 Line 274). We have also added “also” in Page 9 Line 274 and “different and” in Line 276.